# CHNS Modeling for Study and Management of Human–Water Interactions at Multiple Scales

**Kumaraswamy Ponnambalam [1],* and S. Jamshid Mousavi [2]**

[1]  Department of Systems Design Engineering, University of Waterloo, Waterloo, ON N2L 5R5, Canada
[2]  School of Petroleum, Civil and Mining Engineering, Amirkabir University of Technology
    (Tehran Polytechnic), Tehran 15875-4413, Iran; jmosavi@aut.ac.ir
*  Correspondence: ponnu@uwaterloo.ca

**Abstract:** This paper presents basic definitions and challenges/opportunities from different perspectives to study and control water cycle impacts on society and vice versa. The wider and increased interactions and their consequences such as global warming and climate change, and the role of complex institutional- and governance-related socioeconomic-environmental issues bring forth new challenges. Hydrology and integrated water resources management (IWRM from the viewpoint of an engineering planner) do not exclude in their scopes the study of the impact of changes in global hydrology from societal actions and their feedback effects on the local/global hydrology. However, it is useful to have unique emphasis through specialized fields such as hydrosociology (including the society in planning water projects, from the viewpoint of the humanities) and sociohydrology (recognizing the large-scale impacts society has on hydrology, from the viewpoint of science). Global hydrological models have been developed for large-scale hydrology with few parameters to calibrate at local scale, and integrated assessment models have been developed for multiple sectors including water. It is important not to do these studies with a silo mindset, as problems in water and society require highly interdisciplinary skills, but flexibility and acceptance of diverse views will progress these studies and their usefulness to society. To deal with complexities in water and society, systems modeling is likely the only practical approach and is the viewpoint of researchers using coupled human–natural systems (CHNS) models. The focus and the novelty in this paper is to clarify some of these challenges faced in CHNS modeling, such as spatiotemporal scale variations, scaling issues, institutional issues, and suggestions for appropriate mathematical tools for dealing with these issues.

**Keywords:** coupled human–natural systems; integrated water resources management; sociohydrology; modeling perspectives; agent-based modeling; differential equations; system dynamics; uncertainty; artificial intelligence; machine learning

---

## 1. Introduction

*"By the continuance of rain the world is preserved in existence; it is therefore worthy to be called ambrosia"*, Thirukkural [1]—Couplet 11 and *"Even the wealth of the wide sea will be diminished, if the cloud that has drawn (its waters) up gives them not back again (in rain)"*, Thirukkural [1]— Couplet 17, (From about 2000 years ago).

Hydrology is defined [2] as "the science which deals with the waters of the earth, their occurrence, circulation and distribution on the planet, their physical and chemical properties and their interactions with the physical and biological environment, including their responses to human activity". Integrated water resources management (IWRM) is defined as "a process which promotes

the coordinated development and management of water, land and related resources, in order to maximize the resultant economic and social welfare in an equitable manner without compromising the sustainability of vital ecosystems" [3]. By these definitions, hydrology is a descriptive tool and IWRM is a prescriptive tool which, by necessity, depends on descriptive tools and must be interdisciplinary. Choi and Pak [4] clarified that "Interdisciplinarity analyzes, synthesizes and harmonizes links between disciplines into a coordinated and coherent whole". While this is a very difficult requirement, some humble attempts to define interdisciplinary approaches have been made through these proposed fields: hydrosociology was motivated by Falkenmark [5] to encourage developing a field to include, from the humanities viewpoint, the society in the planning of water projects, and Sivapalan et al. [6] introduced sociohydrology (SH) as a new emerging branch of hydrology from a scientific viewpoint focusing on sociohydrologic systems as coupled human–natural (CHNS) systems including the feedback on to both local and global systems, where dynamics of the two systems and their co-evolution should be explicitly accounted for.

These interdisciplinary approaches require tools in order to solve real world problems. A decade before presenting sociohydrology (SH), system dynamics (SD) models, an important tool of systems modelers, was used by Simonovic [7] under the umbrella of integrated modeling to consider coupled human–natural systems in his WorldWater model, although not considering the important feedback of evapotranspirations from land and oceans to precipitation which SH proposes to address. It is noted that SD should not be restricted to any particular mathematical construct but they are only lumped systems models and hence are suitable for global level modeling of a large number of systems. SH has been promoting applications of CHNS, and several works have been contributing to SH from both conceptual frameworks and modeling approaches perspectives (e.g., Di Baldassarre et al. [8], Sivapalan and Blöschl [9], Blair and Buytaert [10], Sivapalan and Blöschl [11], Di Baldassarre et al. [12], Xi-Liu and Qing-Xian [13], and Di Baldassarre et al. [14]).

Xu et al. [15] argued eloquently for including social elements in CHNS models. Wesselink et al. [16] summarized well the differences between SH and hydrosociology (HS), and, although we do not discuss this here in detail, they considered the main differences between SH and HS as descriptive versus critical, objective versus subjective, and nature centric versus society centric, among others. SH aims at addressing dynamic cross-scale interactions and feedbacks between natural and human processes that can cause water sustainability challenges [17]. Additionally, there are arguments on whether or not SH can be considered as a new discipline; for example, Sivakumar [18] and Koutsoyiannis [19] who in spite of Wesselink et al. [16] are not convinced that SH has substantial differences with hydrosociology or is a new science, respectively.

The research cited thus far comes from water specialists, but others such as ecologists and economists [20,21] as well as the USA National Science Foundation have also recognized the importance of studying CHNS. Fu and Wei [22] stated "Humans as a group have learned numerous unpleasant lessons for keeping fit for changes in coupled natural and human systems (CNH). . . . Furthermore, . . . we have a limited understanding of the dynamic mechanisms of CNH; therefore, we have been unable to provide a manual for humans' ability to keep fit for a more sustainable global environment."

More than two hundred hydrologists all around the world contributed to a specific paper [23], reporting the systematic procedure taken for identifying 23 unsolved questions in hydrology to streamline future research in this field, the same as what David Hilbert did in 1900 for mathematics. The problems identified are mainly about understanding how change propagates across interfaces within the hydrological system and across disciplinary boundaries, and in particular human interactions with nature and water cycle feedbacks [23]. Among the 23 questions introduced, the following questions are directly related to CHNS, its conceptualization, or suitable modeling tools, implying the importance of studying CHNS in the future of hydrology: "Q4. What are the impacts of land cover change and soil disturbances on water and energy fluxes at the land surface, and on the resulting groundwater recharge?" [23], "Q6. What are the hydrologic laws at the catchment scale and how do they change with scale?", "Q7. Why is most flow preferential across multiple scales and how does such behaviour

co-evolve with the critical zone?" (Q7 in fact addresses the main issues related to the distribution and nature of flow paths raised in Q5 and Q6), "Q18. How can we extract information from available data on human and water systems in order to inform the building process of socio-hydrological models and conceptualisations?", "Q21. How can the (un)certainty in hydrological predictions be communicated to decision makers and the general public?", "Q22. What are the synergies and tradeoffs between societal goals related to water management (e.g., water–environment–energy–food–health)?", and "Q23. What is the role of water in migration, urbanisation and the dynamics of human civilisations, and what are the implications for contemporary water management?" [23].

What has made CHNS analysis more important than before is what has happened to our societies and the Earth system during the last century, especially in the last seven decades. Humans have been utilizing more natural resources of the Earth such as oil, minerals, soil, and particularly water for different uses, affecting strongly ecosystems services. Therefore, we are currently impacting the Earth system's elements, resources, and processes much more than before. These impacts, which used to be more local, have become more intense (locally) and broader (globally). Consequently, the Earth system's response to the huge amount of human interventions has now returned to impact us and our societies. Global warming and climate change and their impacts are just examples for the mentioned influences. Therefore, what we have done on the Earth has come back to affect us in unplanned and unexpected ways; thus, from a system perspective, we are currently facing a coupled system consisting of two main human and natural systems interacting with each other. As a result, we cannot model each of these two systems without accounting for the feedback it receives from the other one. This high coupling makes interdisciplinarity studies a requirement, and CHNS modeling and the quantitative tools for such modeling are inevitable. Several other works making the above points are related to the idea of "planetary boundaries" in references [24,25] or those referring to water scarcity in ref. [26] and water security in ref. [27].

Figure 1 presents symbolic views of interactions between these two systems in the 18th to 20th centuries and the current 21st century view. The top figure is how most hydrology and water management were studied where the people simply depended on the world's global water cycle as an input and did not consider that people were changing it at global scales, which was a reasonable assumption for most of history. In the view now (the bottom one), this interaction between people and global hydrology is explicitly considered as tightly coupled and indicating huge impacts humans have on Earth entering the Anthropocene, thus bringing enormous challenges to modeling and computational research.

Nikolic and Simonovic [29] suggested a generic multi-method modeling framework for support of IWRM to capture structural complexities of water resources systems and to examine the codependence between these systems and socioeconomic environment. Loucks [30] stressed the need for a new kind of water resources planning and management modeling expertise addressing a wider range of societal concerns that stem from the impact water has on human activities. Given the interconnectedness of water and socioeconomic systems, he reminded systems modelers of the need for viewing a water resource system as a coupled social-economic and ecological system and for developing models capable of estimating the possible social impacts, and capturing the adaptive capacity of these systems to learn and innovate in response to change [30]. However, some of the challenges are due to feedbacks between social and natural (water) systems at different spatiotemporal scales. In the last two decades, there have been several attempts for introducing and dealing with CHNS and the associated frameworks, concepts, and modeling tools. Stevenson [31] presented a framework for CHNS consisting of five elements: human well-being, environmental policy, human activities, stressors (contaminants, pollution loads, etc.), and ecosystem services for environmental management problems. Propagation of thresholds in relationships among the elements through CHNS is the key aspect of the proposed framework. Senf et al. [32] used remote sensing as an information source for modeling the central Europe forest ecosystem dynamics and mapping forest disturbances.

Before Anthropocene

Anthropocene Era

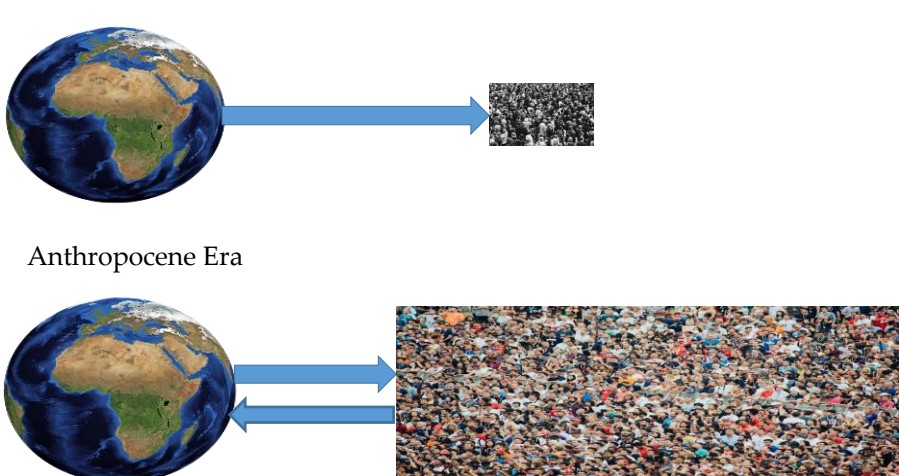

**Figure 1.** Different perspectives of impacts of people on Earth and vice versa in two different time periods (Small crowd picture: https://unsplash.com/photos/sUXXO3xPBYo Globe: https://www.hiclipart.com/free-transparent-background-png-clipart-dxjin/download Large crowd picture: https://unsplash.com/photos/8I423fRMwjM) [28].

Integrated assessment models (IAMs) and global hydrological models (GHMs) are among other modeling approaches attempting to address feedbacks at a global scale. IAMs as complex socioeconomic models incorporate water resources in terms of both supply (simplified hydrology) and demand (water use) in references [33,34]. IAMs are lumped systems models that use the global change assessment model (GCAM). GCAM [35], being under development for over 35 years, is a global model that represents the behavior of, and interactions between, five systems: the energy system, water, agriculture and land use, the economy, and the climate. These lumped system models have been considered as one of the tools of CHNS in ref. [36]. On the other hand, GHMs incorporate water demand scenarios into complex hydrological models although adding genuine feedbacks is ongoing. GHMs distinguish their approach from other global hydrological models by having a few parameters to calibrate especially at the local scale. Any parameter estimation it does is done at a large scale such as at the ecoregion, large river basins or climatic regions (see, e.g., the review paper by Sood and Smakhtin [37]). GHMs run in a grid format at a spatial resolution of 0.5 degrees (just over 3100 km$^2$ per grid cell at the Equator) at worse and temporal resolution of a day. Of course, the requirement of data to fit this spatiotemporal scale for the entire globe makes it harder to apply uniformly leading to many different models. The match between various GHMs is normally not that good and hence many uncertainties in their results are yet to be fully understood. Brêda et al. [38] presented a recent GHM application in South America indicating that, without the ability to calibrate, the results are difficult to judge. PCR-GLOBWB 2 is also a GHM that is worth noting as it has 5 arc-min resolution, about 10 km at the equator, and can optionally couple with other well-known models, e.g., MODFLOW [39].

CHNS modeling has been considered in IWRM, complex water resources systems analysis, and water–energy–food–environment nexus approach using different systems analysis simulation and optimization approaches. For instance, Cai et al. [40] formulated a basin-scale integrated hydrologic–agronomic–economic model as a highly nonlinear mathematical program. In particular, the SD method and its advantage in modeling nonlinear feedbacks was employed to simulate interactions between social and natural systems of complex water resources systems. Simonovic's [7] WorldWater SD-based model integrates water resources sector with five driving sectors of industrial growth: population, agriculture, economy, nonrenewable resources, and persistent pollution. The model results demonstrate the strong relationship between the water resources and industrial

growth. Simonovic and Davies [41] stated "the current approach to understanding connections between biophysical and socioeconomic systems requires their artificial separation via modelling techniques. Such an approach explicitly excludes those feedbacks critical to understanding the behaviour of climate and socioeconomic systems". Subsequently, they discussed the unreal simplifying assumptions made because of excluding the feedbacks that operate between biophysical and socioeconomic systems. Prodanovic and Simonovic [42] coupled a continuous hydrologic model and a socioeconomic model using SD, where the hydrologic component responds to changes in socioeconomic conditions, and socioeconomic conditions are influenced by hydrologic quantities. Davies and Simonovic [43] pointed out that water resources models have traditionally considered socioeconomic and environmental changes as external drivers, thus they have mainly focused on water systems. Therefore, to provide insight into the nature and structure of connections between water resources and socioeconomic and environmental changes, they presented a SD-based integrated assessment model incorporating dynamic representations of these systems through nonlinear feedbacks. Lam et al. [44] proposed a modeling framework to analyze the sustainability problem in Mississippi River Delta. The framework includes six components from the natural and human systems linked together, and the SD approach is used to model the feedback loops among the components. There are more recent SD applications in water and hydrologic systems in ref. [45], which is a review article, and in ref. [46], which presents a watershed scale application.

Agent-based modeling (ABM) have been utilized as one of the main quantitative techniques for CHNS analysis [47]. Walker et al. [48] offered a conceptual model of a coupled social–water system and proposed analytic approaches to support policymaking for environmental and water resources planning and management. Dziubanski et al. [49] built a sociohydrological model by integrating ABM and a quasi-distributed hydrologic model, in which the impacts of land-cover changes resulting from decisions made by two different agent types are simulated using the curve number method. The model is used to simulate scenarios of crop yields, crop prices, and conservation subsidies considering varied farmer parameters representing the effects of human system variables on peak discharges. Noël and Cai [50] focused on how to quantify the role of individuals in models of CHNS in a basin-scale irrigation management problem, where the human sub-system is a community of farmers. They coupled an agent-based model, simulating farmers' behavior, and a groundwater model and concluded that such behavior can be considered as an additional source of uncertainty in the CHNS model proposed. ABM has also been used for systematically studying interactions among hydrology, climate, and strategic human decision making in a watershed system [51] and among those mentioned elements and landscape-scale forest ecosystems [52] as CHNS.

Pouladi et al. [53] presented a sociohydrological modeling framework for complex water resources systems performance assessment by combining ABM and the theory of planned behavior (TPB) to account for farmers' behavior in the Lake Urmia Basin, Iran. The framework was then extended by Pouladi et al. [54], who integrated ABM and data mining for capturing farmers' sociohydrological interactions and complex behavior in response to drought conditions. They employed the association rule to discover the main patterns from the field data collected, representing the farmers' agricultural decisions. The rules discovered were used then in ABM as the behavioral rules to simulate the agricultural activities. Aghaie et al. [55] presented an agent-based groundwater market model to analyze the economic and hydrologic impacts of different market mechanisms and water buyback programs.

Based on above review of literature, different branches of the Earth system science, IWRM, and SH have been emphasizing the need for CHNS modeling. In the following section, we elaborate further on overlapping and distinctive aspects of IWRM and SH, both of which have contributed to the progress and advancement of CHNS analysis. Then, as part of our aim in this paper, we discuss on how systems analysis approaches, e.g., SD, ABM, stochastic differential equations, and optimization, along with artificial intelligence (AI), machine learning (ML), and data analytics algorithms, are used in modeling and quantifying co-ordination and integration as the heart of IWRM definition and co-evolution stressed in SH, respectively.

## 2. Materials and Methods

### 2.1. IWRM and Sociohydrology (SH)

Before presenting our view on similarities and differences between IWRM and SH, we provide some other important views on this matter as mentioned in the 23-unsolved-problems-in-hydrology paper where it is stated "The traditional support that hydrology has provided to water resources management [56] in its dual role of (i) quantifying hydrological extremes and resources relative to societal needs and (ii) quantifying the impact society has on the water cycle, is now broadened in a number of ways. First, more integrated questions of the long-term dynamic feedbacks between the natural, technical and social dimensions of human-water systems. While water resources systems analysis [57] has dealt with such interactions from an optimisation perspective on a case-by-case basis, much is to be learned by developing a general understanding of phenomena that arise from the interactions between water and human systems. Thus, as socio-economic perspectives [58,59] are being integrated in these feedbacks, the interest is not only on decision support but also on the role of society in the hydrological cycle in its own right. Second, . . . , the topic of water and health (e.g., Mayer et al. [60] and Dingemans et al. [61]), as well as spatial problems such as the interaction of migration and water issues. Third, . . . Also, water is traded globally through the water–energy–food nexus, and it will be interesting to see what role hydrology can play in this nexus [62]." (Blöschl et al. [23] (p. 1152)).

Several important issues are highlighted in GWP's (2000) IWRM definition that it is "a process which promotes . . . " [3] such as managing resources in a coordinated way, considering other related resources than just water (soil, ecosystem, etc.), social welfare in addition to economic objectives, equity, sustainability, and environment. However, to apply these important issues in water management practices, it is also necessary to know what kind of science, approaches, and modeling tools we need to help promote the above process. For example, Metz and Glaus [63] mentioned that the integration of water-related policies in IWRM is challenging because policy actors should coordinate their demands and actions across policy sectors, territorial entities, and decision-making levels within a water basin, whereas actors are restricted by the policy framework. Biswas [64] provided a critique of IWRM in practice, but, given the comprehensiveness required by its definition, it is fair that most studies could not satisfy the definition.

Fu and Wei [22] presented the conceptual cascade of "pattern–process–service (function)–sustainability" for understanding of diagnoses and practices for keeping fit in CHNS. They stated "The former refers to understanding the dynamics of CHNS, and the latter refers to management policies and practices for improving sustainability." Therefore, from such a perspective, SH is more related to the diagnostic understanding, whereas IWRM is more concerned with the practices needed for the keeping-fit concept in CHNS. It is worth noting that "keep fit" is considered as the process of matching a socioeconomic system with its biophysical environment across temporal and spatial scales, while bidirectional coupling exists between environmental changes and socioeconomic changes [22].

The concept of IWRM moves away from top-down "water master planning" and toward "comprehensive water policy planning". The IWRM concept already recognizes the role and importance of two other subsectors, socioeconomic and institutional, and then the natural subsector and addresses the interactions between these subsectors as described well in ref. [65].

IWRM has been mainly concerned with management aspects of technical, socioeconomic, and institutional dimensions of water-related decision making. SH has been proposed for studying water–human systems dynamics and emphasizes the consideration of the co-evolution of natural (hydrologic) and social systems, and to consider human systems feedback on global water cycle and vice versa. Therefore, explicit considerations of multiscale feedbacks proposed in SH is useful for a comprehensive application of IWRM, which the CHNS modeling approach facilitates.

The importance of scale-related challenges has been recognized in SH. For example, Blöschl et al. [23] (p. 1152) stated "The challenges lie in linking short-term local processes (what

we have mostly studied in the past) to long-term global processes, and vice versa". In terms of temporal scale, both IWRM and SH can address different time scales, especially long-term impacts under notions of sustainability and (time) evolution, respectively. However, the co-evolution of human–natural systems in SH is more relevant to, and compatible with, the need for considering nonstationarity aspects in these complex systems than tools and methods being developed and used in IWRM. Regarding the spatial scale, as a step ahead and moving from a single project-scale approach to a wider spatial-scale modeling framework, IWRM has mainly focused on basin- or regional-scale analyses, thus IWRM considers the impacts of human interventions on water budget components locally at a basin scale. Therefore, human interventions such as urbanization effects, dam constructions, expansion of irrigated areas, etc., and their impacts on local (basin scale) water cycle have already been recognized by current water resources management practices. On the other hand, the impact of global water cycle on local basin-scale water resources systems and societies when considered takes a typical one-way scenario-based approach, with few exceptions (see ref. [66]). Nevertheless, the cumulative impacts of the mentioned human interventions on global water cycle goes far beyond a basin-scale approach. In other words, current hydrologic and water-resource management approaches do not sufficiently account for the impact of basin-scale human-induced water resource-related activities on the large-scale hydrology of Earth, and many long-term hydrologic predictions do not account for global forces that influence local sources and vice versa, something that SH is proposing to address. Van der Ent et al.'s [67] study of moisture tracking on a global scale is a good example of such suggested studies in SH.

In response to the above-mentioned points, SH focuses on some other important issues lacking in well-established science of hydrology and IWRM such as impacts of human interventions on global water cycle directly and explicitly. By "directly and explicitly", we mean SH highlights the importance of considering human (social) and natural (hydrologic) systems as co-evolving coupled systems, whereas their interactions are considered as part of the systems itself, through for example a two-way feedback approach, not just via a one-way scenario-based approach. To do so, SH requires quantifying approaches and modeling frameworks, and CHNS modeling is a possible framework. When studying CHNS, we embed human–natural systems including water. From such a point of view, SH through CHNS becomes a study of complex system of systems, a type of system discussed well for example by Haimes [68].

Let us clarify what we explain above about local- and global-scale impacts through a simple well-known problem in classical hydrology. Urbanization is a good, known example of a human (social) system impact on the hydrologic cycle locally. Local-scale effects of urbanization, especially flooding issues, are typically considered through designing a proper urban drainage system. The impacts are increases in surface runoff; reductions in infiltration, groundwater recharge, and evaporation, degradation of water quality indices, etc. In response to the impacts of such a socioeconomic-driven intervention on local water cycle components, engineers design proper drainage systems and wastewater treatment plants to efficiently manage stormwater resulting from land use change and increased impervious lands. Additionally, comprehensive integrated models are currently available for assessing a longer than normal time horizon (e.g., 100 years) impact of urbanization on a basin, similar to the work of Luo et al. [69], who studied the loss of land from agriculture and forestry to urban areas to increase land runoff to over 60% in China. However, what about the impacts of these changes on the global water cycle components and our responses to them? What is the cumulative impact of millions of small to large developed urbanized areas in different basins, countries, regions, and continents all around the world on the global water cycle? Have we addressed this latter-type impact adequately in our hydrologic and water resources models? We believe this is something SH has correctly attempted to address. In this line, among 23 unsolved problems in hydrology, Question 23 particularly emphasizes migration and urbanization as key topics to focus on in human–water interactions [23].

A point to mention, however, is that, although CHNS has mainly been emphasized in SH, we believe CHNS modeling tools, especially feedback-based socioeconomic–natural systems modeling, has already been used in IWRM. In other words, the authors of this paper think that, while the concept of co-evolving coupled human–natural systems emphasized in SH is essential and of great importance, the SH community may not have adequately recognized several valuable attempts accounting explicitly for coupled human–natural systems modeling done under names and areas other than SH. The attempts have particularly taken place using system dynamics (SD) approach in IWRM, systems approach to complex water resources systems, and water–energy–food–environment nexus approach, some of which were cited in the Introduction (e.g., the works of Simonovic [7], Simonovic and Davies [41], Prodanovic and Simonovic [42], Davies and Simonovic [43], Walker et al. [48], Noël and Cai [50], and Loucks [30]). Additionally, SH does not explicitly talk about the role and importance of institutions and institutional subsystem underlying the co-evolution of human (social) and natural systems, something recognized by IWRM at least conceptually, if not quantitatively. This issue is explained further and discussed later in Section 2.2.4.

The co-evolutionary thinking of SH will be of help and importance in the current and the future of hydrology as a science. However, there is still a significant gap in developing and advancing computational CHNS modeling approaches under any of SH, IWRM, or food–energy–water–environment–health nexus contexts. A tutorial was presented, including as many of the modern computational approaches necessary for considering such challenges in CHNS, by Ponnambalam et al. [28].

It is worth mentioning that, despite the significance of CHNS analysis, there are not as many works that have successfully applied quantitative approaches and models of CHNS as there should be, mainly because of the complexity level of the models required for tackling such a task. In this regard, Loucks [70] stated "Simplification is why we model . . . We know that our simplified models will be wrong. But, we develop them because they can be useful. The simpler and hence the more understandable models are the more likely they will be useful, and used, as long as they do the job." Therefore, he raised an important question: " . . . what level of model complexity is needed to do a job when the information needs of that job are uncertain and changing?" For instance, we typically assume in IWRM that future water availability and demand values are known or can be estimated (in deterministic or probabilistic sense) without explicitly accounting for the impacts of socioinstitutional systems on their future estimations. However, the point is how challenging the consideration of such impacts would be, and what assumptions, information, and modeling techniques would be needed to overcome the complexities it brings. In this regard, Walker et al. [48] suggested that environmental and water systems decision makers need to consider social responses as well as economic and environmental impacts of their decisions, but predictions of such responses will not be accurate, especially in the future, and hence requiring uncertainty modeling and its inclusion, another thing not well done. To better understand the coupled social and natural components of water resource systems, they then provide some examples of how hard it is to attempt predictions, why, and the consequences if those predictions are wrong.

In the following, we elaborate more on CHNS modeling challenges. In addition, opportunities and potential modeling tools that can address some of these challenges are introduced and discussed.

## 2.2. CHNS Modeling: Challenges and Opportunities

It is worth noting that SH emphasizes co-evolution to be considered in CHNS modeling. However, from a quantitative and mathematical modeling perspective, it would be very hard and challenging to fully model all behavioral aspects of these very complex coupled systems. Below, we explain and discuss our view on different aspects of such challenges and the complexities involved and provide links to tools:

2.2.1. Mismatch in Temporal Scale and Time Resolution

We first discuss the mismatch in temporal scale and time resolution of models and variables of interest in social systems and those in natural systems. Such a challenge arises while trying to answer questions such as "What is the role of water in migration, urbanisation and the dynamics of human civilisations, and what are the implications for contemporary water management?" as the 23rd question mentioned by Blöschl et al. [23]. To clarify such an inconsistency, suppose a basin-scale water allocation model is the one in which we are going to simulate both natural (physical) processes and social processes and their interrelationships. Governing equations and variables in the natural (physical) system are mass balance equations over time and space or other hydrologic-related equations, reservoir releases, water allocations to demand sites, etc. at a certain temporal scale. On the other hand, governing equations and variables in the socioeconomic system are those related to processes of poverty, migration, income, education, gender equity, etc. at another temporal scale. The type of related equations for the socioeconomic system could be statistical regression equations derived from analysis of corresponding data and information collected from questionnaire or supply and demand curves and functions fitted to data, where societal variables and signals of interest are the number of migrated people from and different sites, average household income, etc.

Traditional natural system-focused water allocation models solve water balance equations over time and space along with operation rules and policies on how to make releases from the reservoirs under predefined demand satisfaction priorities. The corresponding water allocation problem is typically formulated as single- or multi-period network flow programs (NFPs) or linear programs (LPs) solved iteratively by fast out-of-kilter, Lagrangian relaxation, or dual Simplex algorithms. In this approach, socioeconomic considerations are only accounted for indirectly through values of water demands and the priority numbers selected for different demand types and sites and target reservoir levels, which are determined outside the model. What is directly considered inside the model is the mentioned equations related to physical (natural) subsystem. Such a framework is currently being employed in well-known, well-established river basin decision support systems (DSSs) such as ModSim and WEAP.

The question here is: What challenges do we have if we want to develop a new coupled human–natural, sociohydrologic-based water allocation model integrating the mentioned natural system-focused water allocation DSS and a socioeconomic model at varying temporal scales? Such a coupled model aims to simulate both physical and social processes and variables of interest and their interrelations and impacts on each other. This is because we know that more reliable, timely, and adequate water allocations from both water quantity and quality aspects can improve societal and economic signals and indicators in that site or region. On the other hand, a demand area having better welfare indices motivates more development, as well as population and economic growth, resulting in higher levels of water demand, which will therefore put more stress on natural hydrologic system when the quantity of water resources is limited. One rising challenge here is that the typical time resolution considered for (hydrologic) variables of the natural system-focused water allocation model, e.g., water allocation values and reservoir releases, is one day to one month in duration.

Suppose that we are able to develop a quantitative socioeconomic model, simulating social processes and variable of interest, e.g., poverty, income, migration, education quality and level, health, etc., to couple it with the water allocation model. These variables and processes would respond to the changes of water allocation variables on a much longer time interval basis than a month, usually in years of interval. It may not be reasonable to have a monthly-basis socioeconomic model; instead, a model with at least a yearly time step may be more meaningful. Therefore, models of natural systems (physics-based, conceptual, etc.) could be of hourly/daily/monthly basis, while such time resolutions may not be suitable to be selected as the time resolution of societal signals quantifying processes of poverty, unemployment, education level, income, migration, etc. This means that we need to couple two models with two different time resolutions requiring specific considerations from modeling point of view.

To deal with such a challenge generally in CHNS with multiple time resolutions representing different slow to fast natural or social processes concurrently, stiff ordinary differential equations (ODEs) could provide an opportunity and framework. Stiff ODEs are appropriate tools when two or more temporal scales are involved as faced here in coevolution. In addition, these ODEs may have to be stochastic ODEs due to various noisy processes encountered. These problems have been anticipated by many, including Sivapalan and Blöschl [9], but we propose appropriate mathematical solutions for such problems here and in the forthcoming tutorial paper based on Ponnambalam et al.' [28] workshop.

2.2.2. Mismatch in Type of Models and Modeling Approaches

The functions and relationships simulating physical processes in natural systems may form partial deferential equations (PDEs) of conservation of mass, momentum, and energy based on Newtonian physics. Conceptual hydrologic models are also developed on the same basis with different levels of approximations often as lumped systems (ODEs) in modeling the physics of the processes of interest. Then, approximations and uncertainties in input, model structure, and model parameters are accounted for through error-minimizing calibration approaches of these hydrologic models. Of course, applications of data-driven models approximating physics-based and conceptual hydrologic models and processes using model-free artificial intelligence (AI) and machine learning (ML) algorithms have been advancing during the last three decades. This is still an ongoing emerging field in modeling a wide range of analysis and design problems in engineering systems. Similarly, there are physics-based models that are able to simulate physics of macro- or micro-economic processes, e.g., behavior of suppliers and consumers under different market conditions. However, this may not be the case for social processes, i.e., poverty, health, gender equity, migration, education level, etc. In other words, social systems and processes, despite being complex, are being modeled by PDEs or other well established traditional mathematical approaches and equations, but the research is still in early stages for applications in the real world [71].

Traditional mathematics, although very powerful, still is not able to fully capture the physics of these processes and the related disorganized complex social systems. That is why it is said that the social systems are the most complex systems. Future advances in sociology, psychology, and mathematics and interactions and co-operations between sociologists, psychologists, and mathematicians may make it possible in the future to come up with a fuzzy or stochastic PDE solution of which simulates the time evolution of a society. However, up to now, and considering the current available modeling technology, models simulating social processes could be either qualitative or, in the best case, of empirical, statistical type. These empirical models are typically developed based on data and information collected on the variables of interest over time and space using questionnaires encapsulating experts' judgements and knowledge or other socioeconomic data collection and monitoring systems. Consequently, the type of qualitative or quasi-mathematical, empirical models simulating social processes will be different from quantitative mathematical models simulating natural systems. In other words, CHNS modeling requires integration of normative/quantitative, physics/mathematics-based hydro-economic models and subjective, qualitative human mimicking- or data-driven socioeconomic models.

The above-mentioned requirement calls for specific considerations and involves additional complexities when it comes to their calibration and verification and other modeling aspects. For example, model-free AI/ML-based methods with no explicitly defined analytical expressions and functions for quantifying relationships among social variables would restrict putting them among the set of constraints of fast gradient-based network flow programming (NFP) or linear programming (LP) algorithms being used in current basin-scale water allocation DSSs such as ModSim and WEAP. In such a situation, evolutionary optimization algorithms (EOA) that are much slower than NFP and LP would be the only choice to be employed as optimizers in the mentioned DSSs. This is also the case if the model wants to be put into an optimization framework optimizing reservoir releases, water allocations, and other socioeconomic decision variables of interest. Although EOA are very useful, they would slow down the convergence rate of the resulting optimization algorithm when

many more function evaluations are required. Therefore, it would be inevitable to think of applying other meta-model-based optimization algorithms. If any single run of the coupled human–natural water allocation model is computationally intensive, then meta-modeling and surrogate optimization will be beneficial to speed up the computation power (Mousavi and Shourian [72], Kamali et al. [73], Mirfenderesgi and Mousavi [74]).

Apart from the above-mentioned challenges and opportunities, there is a great potential for applying fuzzy logic and computations as a branch of AI, introduced systematically by Lofi A. Zadeh in 1964 and less specifically by Luckashewics earlier under multivalued logic context. Fuzzy logic has proved its potential and promise in modeling approximate reasoning as a character of human beings and how they think, behave, and react that can help with social decisions not easily modeled by traditional mathematics. The body of literature on applications of fuzzy logic in water resources systems analysis and hydrology is quite rich and well-established, thus we do not refer to them herein. Instead, we focus on discussing its potential in modeling sociohydrologic processes.

In fuzzy inference systems (FISs), societal variables can be considered as linguistic variables whose values are words such as low, medium, and large as fuzzy numbers, rather than crisp exact numbers. Inexact, nonlinear relationships among these variables are represented by a set of fuzzy if–then rules called a fuzzy rule base. Then, all these linguistic variables and fuzzy rules are put into an inference mechanism, e.g., Mamdani and Takeshi–Sugeno FIS, using compositional inference rules. Such a framework has good potential to be used in modeling social systems and processes since in many instances societal signals and their interrelationships cannot be represented by exact, crisp variables and functions, whereas linguistic variables and fuzzy if–then rules enables us to make use of qualitative information about societal variables and their relationships.

FISs are also beneficial where there is no extensive data available. AI and ML algorithms [75] trained with Big Data are model free, data driven powerful tools with their ability to infer very complex relationships underlying these processes. They can extract and infer governing equations and relationships in complex systems from the data without having any knowledge about physical understanding of the relationships. On the other hand, advances in automated data monitoring, collection, and storage systems have provided a great potential for availability of big databases of socioeconomic variables and their spatiotemporal distributions, e.g., infrastructures locations and characteristics, population growth rates and other related variables, land use, soil type, cropping patterns and areas, crop yields and prices, etc. These data can be stored as different layers of a geographic information system (GIS) that can easily be retrieved whenever required. Therefore, recently-developed deep learning algorithms can be utilized to deal with these big databases of socioeconomic variables. Then, the databases and associated AI-based models trained and validated using a large amount of data can be integrated by other physics-based hydrologic models in the model base of a spatial DSS. Note that with available software and hardware technology, DSSs consisting of a model base, a spatial database, and a graphical user interface connected to each other have already provided powerful computerized framework in which different physics-based, conceptual, and data-driven models communicate different large amounts and types of data and information among each other at different levels of complexity or simplifications (modularity). RiverWare [76], MikeBasin [77], GeoDSS [78], and WEAP [79] are examples for such DSSs among several computerized DSSs developed for water resources and hydrologic systems modeling at different spatiotemporal scales.

Another potential opportunity to deal with the mentioned challenges is agent-based modeling (ABM), which can approximate the system-wide behavior of a complex system from individual level behavior of many elements interacting with each other through simple rules. ABM can simulate the emergent behavior of a system from autonomous individual behaviors necessary for modeling socioeconomic processes. It has already been used in modeling farmers' responses to reduced amounts of water allocations during dry periods, when agriculture systems undergo more severe water scarcity. In this line, there are a number of recently-done or ongoing research works attempting to couple agent-based models, simulating farmers' and food growers' reactions (water consumers),

and physics-based water allocation models (see, e.g., Noël and Cai [50], Pouladi et al. [53] Pouladi et al. [54], Aghaie et al. [55]).

As an example problem, we proposed an integrated decentralized reservoir operation optimization model benefiting from ABM for optimal operations of Bukan Dam constructed on Zarineh-Rud River in Lake Urmia (LU), Iran, the second largest salt water lake in the world which is drying out [28]. Considering current critical situation of LU, Bukan Dam operations play a significant role in supplying the environmental water needed for the lake and its restoration plan. That is why planning a concise operational model for water allocations to both agricultural demands and the lake as a vital ecosystem is of crucial importance. In the proposed framework, ABM is one module of a hydro-agro-socioeconomic water allocation scheme embedded in multi-objective optimization to account for farmers' response and behavior against different scenarios of water allocations. A first version of the proposed framework focusing on the supply part of the model was presented by Ponnambalam et al. [28].

These types of coupled models have been developing from about a decade ago as a step towards CHNS modeling in SH. Earlier than that, and apart from the terminology used, the system dynamics (SD) approach has been successfully developing and applied for about two decades in integrated modeling of socioeconomic processes and in lumped hydrologic models. SD-type models in IWRM have paid specific attention to feedback of natural subsystems of water resource systems on their corresponding socioeconomic systems and vice-versa (e.g., Simonovic and Davies [41], Prodanovic and Simonovic [42]). However, as stated above, these works and attempts have used a systems approach for studying complex water resource systems in IWRM or food–energy–water–environment nexus problems.

### 2.2.3. Mismatch in Spatial Scales and Resolutions of Models

One important issue attracted specific attention in the 23-unsolved-problems-in-hydrology paper is the issue of "Hydrological Change" For example, it is stated in the paper that "the interest no longer resides only in providing scenarios of change (as only a decade ago), but in a rich fabric of experiments, data analysis and modelling approaches geared towards understanding the mechanisms of change." (Blöschl et al. [23] (p. 1152)). However, a challenge in responding to such a requirement is that models quantifying impacts of water cycle on human systems and human systems on local and global water cycle components can have totally different spatial domains and resolutions.

To clarify such a challenge, let us consider the well-established modeling frameworks being used in simulating the impact of climate change on basin-scale water-resource systems, as illustrated in Figure 2, where many important concepts are portrayed. Upscaling (aggregation), downscaling (disaggregation or decomposition), scenario-based analysis, and feedback loops are some of the concepts defined next. Upscaling with a bottom-up approach [80] involves taking an observed or theoretical relationship applicable at the point scale, and altering the relationship so that it is applicable at a larger scale. ABM is perfectly suited for this as individual agents and their actions can be aggregated to provide system level responses. ABM is not only applicable in problems of individual to society level aggregations but also can be equally applied in problems requiring PDEs such as diffusion-advection problems [81]. Thus, ABM is a tool that can be applied to model both lumped and distributed systems.

Downscaling involved taking results from coarse gridded, for example, Global Circulation Models (GCM), to finer resolution at the catchment level both in space and in time. Predictor variables are climatic variables such as specific humidity, total precipitation, convective precipitation, sea level pressure, etc. and soil indices, e.g., total soil moisture, slopes, vegetation indices, etc. Statistical methods including multilinear regression are commonly used (see ref. [82]), as well as the newly proposed combination of physically based statistical models in ref. [83].

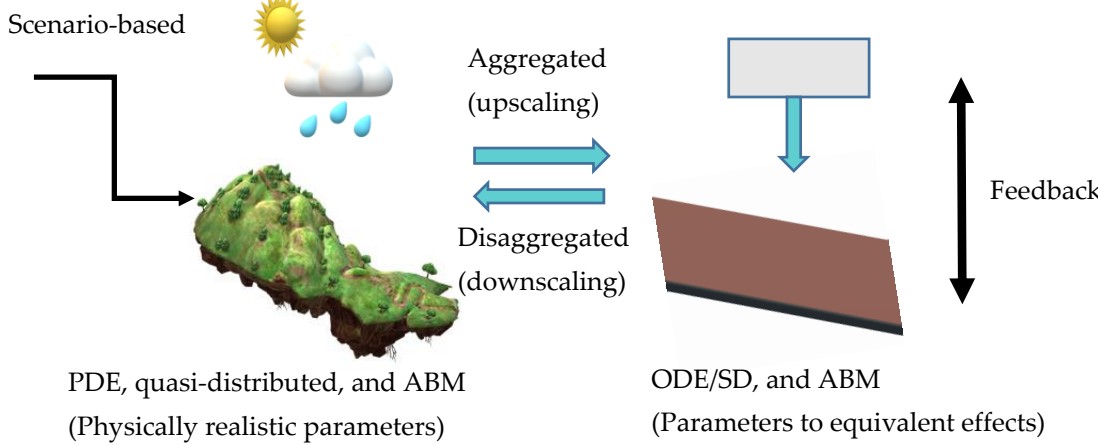

**Figure 2.** An illustration of frameworks for climate change impact assessment studies.

The typical current climate change impact assessment framework is a one-way, scenario-based modeling approach starting with emission scenarios followed by running global circulation models, downscaling, and employing rainfall–runoff models or other basin-wide, natural system-oriented models (the arrow on the left side of Figure 2 indicates this symbolically). However, the final result in terms of future downscaled climate change-driven fluxes of water and energy calculated at a local scale socioeconomic unit do not provide any feedback on emission scenarios and global water. On the other hand, if we imagine there are a tremendous number of basins or regions (local socioeconomic units) impacted by global water cycle through downscaled outcomes of global circulation models, the cumulative impact of all these units on the global water cycle has not yet been accounted for adequately by appropriate upscaled models. This is an impact which is considered important from the sociohydrologic modeling point of view, emphasizing on the co-evolution of the associated CHNS (the bidirectional arrow on the right side of Figure 2 indicates this feedback). However, estimating such feedback from local socioeconomic units on the global water cycle is not an easy task to tackle quantitatively and mathematically. It is because providing feedback from such units to the global water cycle calls for summing effects of all individual-level responses up, something that requires upscaling. This is despite downscaling currently being used in assessing the impact and consequences of global water cycle on much smaller-size spatial socioeconomic units. Even if we can find proper modeling tools and algorithms capable of tackling upscaling such as the global hydrologic models (GHMs) discussed in Section 1 attempting to deal with this issue [37–39], a coupled model equipped by both global-to-local simulation ability (downscaling) and local-to-global modeling capability (upscaling) would become too complex to solve. In other words, it would not be easy to build a fully coupled, two-way local–global model capable of integration (sum) across individual-level behaviors or processes.

We believe ABM having a bottom-up analysis ability to simulate system-level behavior of the global water cycle system through individual-level interactions of basin- or regional-scale socioeconomic units could be an opportunity to be used for the explained challenge of upscaling effects. Additionally, there are studies trying to estimate the contribution of anthropogenic and natural greenhouse emissions to total global greenhouse gas (GHG) emissions and their time variability. For example, using a statistical analysis, Xi-Liu and Qing-Xian [13] estimated the amount of anthropogenic GHG emissions to be about 55% of the global GHG emissions (2016 value). Such works show that we might be able to build a model-free or model-based mechanism simulating anthropogenic and natural GHG emissions as a function of some influencing factors related to hydroclimatic, hydrologic, and societal variables over different spatial regions (terrestrial or ocean systems). We think such simulation models can be put into an ABM approach, summing regional-scales impacts up, and then use it as a GHG emission-generating module in the above-mentioned climate change impact assessment procedure.

If we can do that, the above one-way scenario-based modeling approach would become a CHNS modeling approach.

Regarding other possible opportunities to deal with local-global impact quantification and upscaling, Budyko modeling approaches in ref. [84] are among other possible good opportunities to be used to fill part of the existing gap and challenges in simulating feedback from small-size sociohydrological units to the global water cycle. Additionally, there have been valuable works recently done in hydrologic modeling of evapotranspiration and precipitation simulating impacts of a change in local evaporation realized in a region (point) on other regions over the globe (points). The authors of this paper propose that the output of works, e.g., by Van der Ent et al. [67] and Roy et al. [85], can be represented as a large-scale response matrix of influential coefficients and used in IWRM management models to trace the changes in impacts due to changes in systems which could have happened from policy changes.

Other challenges could be related to mismatch and inconsistency in computational requirements, model precision degrees, and other issues while coupling a detailed model designed for a spatially small-scale unit and a model developed for a much more spatially coarser global-scale spatial unit. Therefore, there will be serious challenges in joint calibration and verification of such a complex coupled model, considering numerical/computational power limitations, inconsistency in their accuracy levels, their discretization scheme over time and space, extent and type of data required for each of them, etc. It seems that it is even impossible to tackle such a task without making several simplifications and assumptions, undermining the positive role of coupling.

Regarding computational burden difficulties in coupling of local- and global-scale models, having different finer and coarser spatial resolutions, two opportunities of parallel (cloud) computing and ML-assisted meta-modeling can be considered. In the first choice, special software and programming settings are used enabling the required computations to be done by several computers in parallel for building a surrogate or meta-model. The meta-model version is much faster than the original model [73]. However, sufficient sample data are required for training a meta-model, and the training procedure can be done online or offline. Therefore, there will be a tradeoff between the approximation power or accuracy of the built meta-model and the number of training samples, each of which requires running the original computationally-intensive model. Balancing these two aspects is an important challenge for doing meta-modeling successfully [72]. These sorts of meta-models have also been used by integrated assessment modeling community in references [33,36].

Another important point and challenge in applying one-way scenario-based approaches while modeling interacting sub-systems is the issue of interfaces between the sub-systems having different scales. The existing traditional approaches consider the issue via boundary conditions to reduce the complexity. However, such a simplifying approach may not be enough in some cases, as pointed out by Blöschl et al. [23] (p. 1152) stating "There is a broad recognition that we need to learn more about interfaces in hydrology. These have traditionally been imposed as boundary conditions, thereby reducing complexity, but we now need to look at the more typical cases where we can and should not do this, as the interfaces couple rather than constrain system behaviour. These interfaces include those between compartments (e.g., atmosphere–vegetation–soil–bedrock–streamflow–hydraulic structures) in three dimensions, interactions between the hydrological fluxes and the media (e.g., soils, vegetation), and interactions between sub-processes that are usually dealt with by different disciplines. (e.g., water chemistry, ecology, soil science, biogeochemistry). Linking these interfaces conceptually and in a quantitative way is currently considered a real bottleneck".

Overall, given the explanations provided, although there are serious challenges mentioned, we have promising tools and algorithms for shifting from the one-way global-to-local approach towards a two-way feedback-based coupled global-to-local, local-to-global modeling approach. For example, Simonovic and Davies [41] discussed other simplifying assumptions made in traditional climate change impact assessment studies including: (1) predictability of the character of all interactions between biophysical and socioeconomic systems, despite their nonlinear nature; (2) irrelevancy of the

interactions between these systems and the behavior of each; and (3) reparability of these two systems so that feedbacks between the systems are external to both. In this line, Davies and Simonovic [66] developed the system dynamic-based ANEMI simulation model for integrated assessment of global change especially the carbon transfers. Additionally, we already referred to a number of numerous sophisticated approaches, including system dynamics and analysis, stochastic simulation and optimization, integrated agro-hydro-socioeconomic modeling, etc., presented by water resources systems analysts and IWRM community before SH's introduction that have gone far beyond scenario based approaches. Koutsoyiannis [19] also stated this fact in his review comments for the first-published SH paper in Hydrological Processes journal.

### 2.2.4. Institutional and Governance-Related Challenges

The co-evolution of human–natural water systems has not been under the impact of de-regulated social systems. Rather, impacts of people and social systems on the water cycle have been taking place under a complex institutional and governance framework and settings that have dynamically been changing over time with significant differences among social units spread over space at a certain time (provinces, states, countries, etc.). This issue has impacted the past and will impact the future's co-evolution of coupled human–natural systems, making CHNS modeling much more complex and difficult to quantify.

Additionally, there is another type of challenges in terms of participating units, institutions, governance, authorities, and co-operations among socio-administrative units of a coupled human–natural system from an IWRM perspective. For instance, under European Water Framework Directive [86], EU Member States have established Coordination and Participation Boards at the river basin level as multi-agency and multi-actor groups, supporting the development of inclusive and coordinated river basin planning and the inclusion of interested parties in decision-making processes [87]. This challenge is also raised when dealing with local–global impacts of water cycle and societies on each other, requiring specific considerations. Blöschl et al. [23] stated "While water governance is limited to the local and national scales, a global perspective is clearly becoming increasingly more important in the context of the UN Agenda 2030 and Sustainable Development Goals, the societal grand challenge of our time [14]."

Water management policies, institutions, and governance at provincial, state, and national levels directly affect basin and regional-scale water cycle components. On the other hand, international-level policies, agreements, rules, regulations, and protocols (e.g., International Law Association Committee on the Uses of the Waters of International Rivers [88], United Nations [89], and United Nations [90]) and commitments established by international organizations and institutions would affect medium- and long-term state of $CO_2$ emission, global warming, and climate change conditions that would influence all local socioeconomic units. For example, basin- and national-level water management practices, policies, and governance (local-scale institutions and governance) underlying socioeconomic activities and interventions (e.g., dam constructions, land-use change, expanded irrigation areas, etc.) combined by climate change impacts (global-scale intervention) during last decades have caused significant destroying consequences on the drying Lake Urmia ecosystem in Iran. Brazilian government policies have been influential on recent huge burning of Amazon forests resulting in a more adverse $CO_2$ emission condition. This means that co-evolution of our future coupled human–natural systems will certainly be under influence of both local (provincial, state, and national) and global (international) water and environment governance-related conditions. By governance we mean all agreements, institutions, rules and regulations, policies, commitments, protocols, etc.

Under the above-mentioned conditions, can we predict the future of the governance underlying future local–global impacts of our water systems and societies? Can we model and quantify the influence of future presidents of countries and their decisions and commitment level to international $CO_2$ agreements, especially those countries that are more responsible in producing $CO_2$ and its emission to atmosphere? Therefore, are our future coupled human–natural systems really predictable under

such complex governmental and institutional conditions? Do we have better modeling framework and conflict resolution techniques than simple scenario- or feedback-based approaches that can account for deep uncertainties in the future from the aspects just explained? These are some relevant questions in SH and CHNS that are too difficult to answer yet. Loucks [91] stated "What we modellers haven't done yet is to figure out how to make our models suggest planning and management options that we haven't thought of before. This would be an especially important feature for integrated water resources planning and management. Integrated implies that our models have included all the links to all the other major components of our social, economic and, if applicable, ecological environments."

Despite the mentioned difficulties, there are still promising tools and ways to deal with them, which of course are not of mathematical modeling type. Public awareness, NGOs, ease of communications through the Internet and social media, etc. all have promising elements facilitating both national- and international-scale participations and collaborations. Therefore, they provide opportunities for hydrologists to convey their message to policy makers and the society [92].

For example, due to the national-level public awareness and request, the government of Iran established a new organization, Lake Urmia Restoration Program (LURP), coordinating all water management activities for restoring the lake, about six years ago. Under the revised participatory water management policies, help and participation of international parties and collaborators such as FAO and JAICA, and better climatic conditions during last five years, the current situation of the lake has relatively become better, and the continuation of procedures forcing the drying of the lake has fortunately stopped. For the case of Amazon forests, international concerns shown even in the latest summit of the leaders of seven industrialized countries helped the Brazilian government take some actions to control the devastating Amazon forests burning events.

We in this section explained and discussed a number of challenges and complexities system's analysts would encounter while modeling CHNS: (1) mismatches in appropriate time resolutions and space domains of social and physical processes if they want to be included in an integrated coupled model simulating both types of processes; (2) inconsistency among type of models, governing equations, and relationships required to quantitatively simulate societal and physical processes taking place in social and natural systems, hard-to-quantify societal variables of interest, and processes impacting the natural (water) systems both locally and globally; (3) global-to-local (downscaling) and local-to-global (upscaling) challenges and how to model and simulate feedbacks the global water cycle receives from the sum of a huge number of local-scale human-driven impacts; (4) computational CHNS modeling challenges encountered while simulating local and global processes and variables with appropriate spatiotemporal scales and resolutions; and (5) complex and almost impossible-to-predict future governmental and institutional systems, at provincial, basin, national, regional, and international levels, affecting the co-evolution of natural and social systems. For each of these challenges, we also proposed some possible modeling approaches that could help modelers deal with the challenges at least partially and may have to depend on simpler models and meta-models [93]. The new AI/ML techniques provide some new opportunities for promising directions to satisfy both speed and accuracy [94].

## 3. Final Remarks

Systems analysis and coupled human–natural systems (CHNS) models provide the practical approach needed for applications both in the descriptive science of sociohydrology (SH) and in the prescriptive method of integrated water resources management (IWRM). Although CHNS is of great importance, the extent we can develop a coupled human–natural system model mathematically is limited and ongoing. It is nearly impossible to account for all the mentioned sources of complexity required by SH and IWRM in CHNS modeling, the coupling levels of local-to-global and global-to-local processes would depend on data availability for model calibration and verification in the presence of uncertainty. Such a capacity is also restricted by the level of understanding of social, economic, institutional, and natural processes, their governing equations/relations, and their dynamics and evolution. These difficulties may often lead in practice to simpler models that can be solvable

(likely after applying the technique of divide and conquer or decomposition/aggregation to manage different scales in time and space), and verifiable in a limited sense, but adaptive.

**Author Contributions:** K.P. and S.J.M. both developed and proposed the main ideas behind this paper, and K.P. substantially contributed to the rewriting of the first draft written by S.J.M. Figures 1 and 2 were composed by K.P. All authors have read and agreed to the published version of the manuscript.

**Funding:** This research received no external funding; however, the first author has received NSERC, Canada funding for much of his multidisciplinary research.

**Acknowledgments:** Conversations with Slobodan Simonovic, Murugesu Sivapalan, Ximing Cai, Tirupati Bolisetti, James Craig, Tirthankar Roy, and Evan Davies are gratefully acknowledged. We also thank Hamed Hamzekhani and Narjes Ghaderi for their help in formatting references. Constructive comments of two reviewers helped us improve the quality of the final version for which we are very much thankful to them.

**Conflicts of Interest:** The authors declare no conflict of interest.

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
