# Peer review of "CHNS Modeling for Study and Management of Human–Water Interactions at Multiple Scales"

_water, doi:10.3390/w12061699_

Round 1
Reviewer 1 Report
The authors discuss connections between socioeconomic and hydrological systems under the general heading of coupled human-natural systems, with a specific focus on the conceptual frameworks and methods associated with sociohydrology (SH) and IWRM. They present what is largely a literature review that contains a number of definitions and challenges in these two areas, as well as some potential approaches (with examples, in many cases) for coupling human and natural systems. Overall, the paper is timely, clear and well-written, and I recommend its publication. The following are comments to improve particular sections of the paper.
Specific comments:
- Line 38: What was hydrology “redefined” from?
- Line 96: Please clarify question 7.
- Lines 104-120: A number of papers have made these points. It would be useful to refer to some of them. Possible references include the idea of “planetary boundaries” (Rockstrom, 2009 or Steffen, 2015), or papers that refer to topics of water scarcity (cf. Hejazi, TFSC, 2014) or water security (cf. Cook and Bakker, 2012).
- Line 136-158: Missing from this list is the set of integrated assessment models (IAM) and global hydrological models (GHM) that attempt to address feedbacks at a global scale. IAMs are complex socio-economic models that have begun to incorporate water resources in terms of both supply (simplified hydrology) and demand (water use). See for example Kim (2016), Climatic Change. GHMs incorporate water demand scenarios into complex hydrological models. I do not think they currently include genuine feedbacks, but they are increasingly complex and detailed. See for example work by the group at PIK (Potsdam Institute), by Y. Wada, and others.
- Lines: 165-185: The list of SD studies here is appropriate, but the references are somewhat older. Perhaps a few more recent studies could be added.
- Section 1: The Introduction is quite long. The authors could consider shortening this section and merging some of the paragraphs into subsequent sections. The section could also state more clearly the aims of the paper. For example, one aim is provided on Line 244 in section 2.1 – it would be better placed in section 1.
- Line 397: Optimization approaches may not be the best option for complex CHNS systems. The definition of “better” can be very subjective. I believe the authors make the same point, but the argument could be clarified here.
- Line 444: “psychologists” is misspelled.
- Line 456-462: Please clarify this argument. Which “specific considerations” would be necessary?
- Line 503: “Can the authors please provide some examples? This is an interesting approach they describe.
- Line 560: I assume “ML” here is “multiple linear”, but ML was previously “machine learning”, I believe. Please clarify.
- Line 576: I believe some GHM groups have attempted to model these dynamics. Perhaps the authors could check?
- Line 620: The IAM community has begun to explore these sorts of “emulators” (or “meta-models”) as well, with some success.
- Line 632: I did not understand this argument fully. Can the authors please clarify?
- Summary table: The paper could possibly benefit from a summary table of some sort. There are many arguments made that are helpful, and a summary table could help to place the key lessons in context. I leave the decision to the authors; this is merely a suggestion.
Author Response
All comments were accepted and modified at the appropriate places.
Thanks a lot for improving and updating this presentation.
Reviewer 2 Report
Excellent review of literature in coupled human- hydrologic systems. Some of the English is awkward but overall very well written. For example, lines 71-74 are a bit confusing to me. Is SH and HS considered similar enough to be the same sub-discipline or not?
Should the quote on Line 697 begin with the word 'What' ? It will make better sense then.
Again, a good discussion of this SH, IWRM, CHNS links and modeling approaches.
Author Response
Thanks for your comments. We added the correction.